# Stress in the Volunteer Caregiver: Human-Centric Technology Can Support Both Caregivers and People with Dementia

**DOI:** 10.3390/medicina56060257

**Published:** 2020-05-26

**Authors:** Barbara Huelat, Sharon T. Pochron

**Affiliations:** 1Healing Design, Alexandria, VA 22314-4337, USA; 2Sustainability Studies Program, School of Marine and Atmospheric Sciences, Stony Brook University, Stony Brook, NY 11794-3352, USA; Sharon.pochron@stonybrook.edu

**Keywords:** trigger, caregiver, non-pharmacologic therapy, technological innovation

## Abstract

*Background and Objectives*: Currently, one in eight people over the age of 65 have dementia, and approximately 75% of caregiving is provided by volunteer family members with little or no training. This study aimed to quantify points of stress for home-based caregivers with the aim of reducing stress for them while concurrently supporting quality of life for the people with dementia whom they cared for. The overreaching purpose was to increase our knowledge of the caregiver stress burden and explore potential technologies and behaviors to ease it. *Materials and Methods:* We interviewed home-based and professional caregivers regarding causes of emotional and physical stress and methods they used to alleviate it. *Results:* This study found that: (1) dementia symptoms created a burden of stress for home-based caregivers primarily in the areas of medication management, memory loss, hygiene care and disruptive behaviors; (2) home-based caregivers identified “finding available resources” as the most important source of stress relief; (3) a minority of home-based caregivers possessed a resource network and knew how to find resources but all professional caregivers were able to find resources and support; (4) home-based caregivers combated dementia symptoms with positive distractions and human touch with little use of technology, since it was mostly unknown; and (5) facility-based caregivers were knowledgeable and readily used dementia-based technology. *Conclusion:* Since professional caregivers have access to technological resources that our home-based caregivers lack, one might logically conclude that we should transfer technology used by professionals to those with dementia. However, great caution needs to be in place before we take that step. Successful technology should address the human experience as home-based caregivers try to use new technologies. Human-centric technology addresses the needs of both people with dementia and the home-based caregiver.

## 1. Introduction

Our world is entering a pivotal crisis: our aged population is growing [1,2], we have a limited number of skilled caregivers [2,3,4], and we have inadequate housing for the aged [5]. Within the United States alone, 5.8 million Americans live with dementia, and each patient requires caregivers and housing [6,7,8].

Dementia is the most expensive disease to treat; the Alzheimer’s Association estimated a cost of $277 billion annually to care for dementia patients [9]. Additionally, with over 400 unique types and a wide range of severity, treating dementia requires a wide range of caregiver skills. The World Health Organization [10] labeled dementia as a global public health priority.

To meet these challenges, researchers and health care communities have focused on keeping people with dementia at home [2,11], which requires caregivers. Some caregivers are highly qualified and work as trained medical clinicians in senior-living facilities, hospitals, and in patients’ homes. Paid homecare caregivers with little or no medical training also provide supportive care in the areas of housekeeping, meal preparation, and assistance in activities of daily living [12,13].

Approximately 75% of caregiving is provided by volunteer family members with little or no training [14], and 34% of the family caregivers are over the age of 65 [9,13]. Volunteer caregivers experience stress, including relational and financial problems, problems combining different tasks, and caregiving intensity [15,16,17,18]. Caregivers usually take responsibility for supporting the person with dementia, and this excludes caregivers from a normal life. Isolation and exhaustion from caregiving may cause depression and loneliness [13]. The strongest predictor of whether a person with dementia is admitted into a facility was the level of stress experienced by the caregiver [19]. According to the Center for Disease Control [9], a total of 44 million Americans are volunteer caregivers, providing 37 billion hours of unpaid care each year.

Some government-based dementia plans focus on teaming innovative assistive technology with an increased emphasis on volunteer support [20]. Technology can support a person with dementia, potentially reducing the stress of the caregiver, but the caregiver must know about the supportive technology and be capable of using it. Using interview data from both professional and volunteer caregivers, this study aimed to quantify points of stress and the needs of home-based caregivers with the aim of reducing stress for them. The overreaching purpose was to increase our knowledge of the caregiver stress burden and explore potential technologies and behaviors to ease it.

## 2. Materials and Methods

This study used questionnaires to interview two types of caregivers—home-based volunteers and facility-based professionals—about stress-inducing behaviors and issues, resource availability, interventive triggers, and types of technology use when caring for people with dementia. Results from the interviews were used to determine the potential for technology to mitigate the stress associated with caregiving. A qualitative, exploratory design based upon hermeneutical methodology was chosen for this study.

### 2.1. Home-Based Caregivers

Home-based caregivers were selected from a recruitment blog on the first author’s website. At the time of the interviews, 15 participants were selected in the Washington DC metro area. Eleven caregivers were actively engaged in caregiving, and four caregivers discussed love ones who were recently deceased. All of their love ones had some form of dementia (see Table 1).

Interviews were conducted using a questionnaire presented in Appendix A. The questionnaire was developed from human-centric criteria identified in Huelat [21]. The questionnaire also integrated Planetree criteria of patient-centered care [22]. These interviews were conducted between February 2018 and October 2018. Interviews were conducted within the caregiver’s home. Preliminary findings were presented at the Planetree 2018 Conference in Boston and the 2019 Environments for Aging Conference in Salt Lake UT, April 2019.

Participants were provide provided oral and written information about the study by the first author. Information on the purpose of the interview, recording, and data treatment was repeated before each interview. Participants were asked to complete the questionnaire prior to the interview. The wording and degree of probing were adjusted to ensure that the participant felt comfortable.

Potential selection bias may have occurred as participants were volunteers responding to a health care design personal website. Respondents were racially diverse, both male and female, primary over 50, insured and economically secure. Given the small group of participants, the researchers recommend that future studies address the needs of larger and more diverse populations.

### 2.2. Professional Caregivers

Six memory care facilities were selected within the United States for this study; these facilities served economically diverse veterans, and some were gender specific. All facilities were certified Planetree model of care or a patient-centered care provider, and each used patient-centered care (PCC) and human-centric care, defined by the International College of Person-Centered Medicine as, “Medicine of the person, for the person, by the person with the person” [22]. Facilities used a “small house” concept [23], designed as a home-like environment with 12–16 private bedrooms, and a separate dining and living room for the household. Social activities and outdoor activities are shared by 2–3 households in a community setting. Additionally, each facility had a specific population, such as skilled nursing, memory care, and hospice.

As described in Table 2, multi-departmental groups of professional caregivers (e.g., nurse managers, physicians, program directors, environmental services director, Planetree coordinator, and administrative staff) completed the questionnaire shown in Appendix A. We based the questionnaire on Planetree criteria for patient-centered care for people with dementia. The interview questions identified resources available to and used by professional caregivers. Interviews were conducted from 2013 through 2019. Interviews were held within the facility meeting room. Participants gave their consent to participate in the interviews. All data have been de-identified and treated confidentially.

### 2.3. Data Collection and Management

The first author had clinical experience with care work for people with dementia within homecare and memory care facilities. She also had programming, design, and construction experience in memory care environments. Therefore, an open and flexible interview guide was developed to give the interviews the style of an open conversation. The interview conversations were opened by letting the participants talk about themselves, their loved one, and the stress they were experiencing in their role of primary caregiver. The interviews with home-based participant also explored their personal perceptions and attitudes toward living at home with a person with dementia.

Consistent with a multiple case study approach, a range of data types were collected from home-based and professional caregivers. All interviews were transcribed successively, followed by initial analysis of each interview as a single text. After all interviews were transcribed, the authors analyzed each interview.

## 3. Results

### 3.1. Drivers of Stress in the Home-Based Caregiver

The interview data shows that some behaviors exhibited by the dementia patient caused more stress in the home-based caregiver than did other behaviors. On a Likert scale of 1–10, where 1 indicated low stress and 10 indicated high stress, prescription management (mean: 7.9 + 2.2), memory loss (mean: 7.8 + 1.1), hygiene (mean: 6.7 + 2.5), mood swings (mean: 6.5 + 2.1), and disruptive behavior (mean: 6.3 + 2.6) all scored above 5 on average (see Figure 1).

**Medication management** may become complex, and when the dementia patient resists medication, aggression and anger can be triggered [24]. Caregiver 2 said, “My wife was on 35 different medications. It took hours for me to organize them and determine what time she needed to take them and under what conditions. I was always worried that I would get it wrong, or that I would miss one.”

**Memory loss** is exhibited by all dementia patients at some level [25,26]. Caregiver 14 said, “Memory loss in my husband was so sad. He often called me his mother and didn’t recognize our daughter. He called her that nice lady.”

**Hygiene**. Early in the disease, the person with dementia often neglects personal care [27]. Caregiver 9 said, “My father-in-law picked up a hand full of his own bowel-movement and handed it to me and asked what he wanted me to do with it.” Caregiver 2 said, “Getting showered, teeth cleaned and dressed was the most difficult task for me and my husband. He argued about every step and really didn’t know what or how to do even the simplest hygiene tasks, and he wouldn’t let me do them.”

**Disruptive behavior** manifests as screaming, kicking, throwing things, becoming violent, becoming argumentative, and being uncooperative. Caregivers reported often feeling at a loss as to how to handle these situations. Caregiver 2 described a situation where her loved one grew angry while riding in the car and tried to jump out. The husband of Caregiver 6 threw his dishes off the table, trying to communicate that he was ready to leave. 

**Mood swings** manifests as an abrupt and apparently unaccountable change of mood. Caregiver 10 said, “I never know who he is. Some days he’ll wake up the same guy I’ve been married to for 50 years and that same afternoon he accuses me of running around with another guy and wants a divorce.”

### 3.2. Resources for Caregivers

We used verbal interview data to list classes of stress relief identified by the home-based caregivers. While the majority (11/15) identified “finding available resources” as the most important source of stress relief, they also expressed the additional desires detailed in Table 3.

Next, we asked the home-based caregivers where they found the resources they had. As shown in Figure 2, a minority of home-based caregivers possessed a resource network—only 30% of them claimed to have found help through a support group, and 38% of them relied on family. Few of them received official training or support from medical teams. 

We asked facility-based participants where they found resources and support. Unlike the volunteer population, all professional caregivers were able to find help. For example, 100% of them found support through their medical teams, and 80% of them were able to access resources online. See Figure 2.

Although most caregivers had some computer experience, few knew how to conduct a meaningful online search for resources or how to interpret the results. Five caregivers tried dementia support groups and found them helpful in identifying resources. Caregiver 6 said, “I found the GPS trackers through my support group.” However, Caregiver 2 expressed that he could not leave his wife alone for even 10 minutes; he asked, “How could he get to a support group?” Caregiver 2 said, “I didn’t even know there were support groups for caregivers of dementia.”

Family also provided a link to resources. Caregiver 8 said, “My daughter found the meds management dispenser online. She could finally manage her medications on her own, and we quit worrying about her forgetting an important med.”

For the home-based caregivers, doctors and medical professionals were seen as having mixed value. Caregiver 6 said, “The medical team ordered oxygen for my husband and just dropped it off at our home—no instructions, and the medical equipment distributer explained he was just the driver and didn’t know how it worked.”

Family doctors tended to direct caregivers to resources and even hospices. Specialized doctors did not. Caregiver 2 said, “The doctors talked down to me and treated me as if I knew nothing about my husband’s symptoms.”

### 3.3. Triggers

The home-based caregivers frequently combated dementia symptoms with positive triggers. For example, home-based caregivers who identified medication management as a major stressor used positive distractions 20% of the time and technology 80% of the time to mitigate the issue. Home-based caregivers who identified memory loss as a major stressor used positive distractions 100% of the time to cope with the issue; they had no other tools. People who identified hygiene as a major stressor used positive distraction 20% of the time and human touch 80% of the time to mitigate the behavior. See Figure 3.

Caregiver 12 said, “My husband got very agitated when I started to make dinner and started going through the contents of the trash and spilling the contents. I started emptying the trash before I started dinner, and he responded by pulling things out the drawers and closet. Finally, I remembered how he enjoyed reading the paper before dinner. I pulled out some old newspaper and gave them to him daily before I started dinner and he stopped his dinnertime disruptive behaviors.”

### 3.4. Awareness of Dementia-Based Technology

When we compare the technology used by professional caregivers to that used by home-based technology, we can see that much of the technology used by the professionals was unknown to the home-based volunteers. For example, out of the technology commonly used by professionals (e.g., technology used by >80% of the professionals), 20% of the home-based volunteers did not know about monitoring; 26% of them did not know about alarms; 13% of them did not know about meds management; 33% did not know about picture phones, and 46% of them did not know about in-home cameras. See Figure 4.

## 4. Discussion

Results from this study show that five issues generated by the dementia patient (disruptive behavior, mood swings, prescription management, memory loss, and hygiene) cause marked stress to the home-based caregiver; all caregivers voiced a desire to have access to informational and technological resources to cope with these issues. Caregivers used very little technology, and indeed knew very little about technology that had the potential to alleviate stress-causing issues, relying instead on positive distractions and human touch.

Results also show that few home-based caregivers had access to informative and technological resources; they were limited by training, time to get training, and medical knowledge. Some caregivers did not know support groups existed; others had no way to attend support groups since they dedicated their time to caring for the dementia patient. Others received medical technology from doctors without knowing how to use it and not knowing how to access training.

Conversely, all professional caregivers had access to resources. They had medical teams to draw on, and they had educated access to the internet. They also had access to technology that was both unused and unknown to the home-based caregivers.

American home-based caregivers volunteer 37 billion hours of unpaid care each year [9]; globally, especially in countries with a large aged population, per capita volunteer hours are likely similar. Caregivers face conflicting demands from family, work, and social activities [13]. The caregivers we interviewed typically experienced increased financial pressure and loss of income, maybe even their job. They often have to navigate complex forms of insurance, bills, and costs of medications. The caregivers we interviewed face physical and emotional health problems such as stress, burnout, insomnia, depression, poor diet, declining health, and even death. Given that caregiver stress is the biggest driver of when a dementia patient will enter an expensive memory care facility [19], it behooves us to find ways to support the home-based caregiver in addition to the dementia patient.

Since professional caregivers have access to technological resources that our home-based caregivers lack access to and actually do not even know about, one might logically conclude that we should transfer the technology used by the professionals to the homes of the dementia patients. The authors of this paper, however, recommend that great caution be in place before we take that step. Most technology developers focus their efforts on the dementia patients; we suggest that successful technology should instead address the needs of both dementia patients and the home-based caregiver [28,29]. Understanding the human experience as home-based caregivers actually try to use new technologies is essential to their successful implementation [30]. Home-based caregivers have expressed concerns that technology be easy to use, accessible, and functional, and that they receive training.

Patient-centered care (PCC) is a sociopsychological approach to health care that recognizes the individuality of the patient in relation to the attitudes and care practices that surround them [31]. PCC has three essential components: effective communication, human relationships, and promotion of the values, needs and desires of patient [32,33]. Studies indicate that Planetree’s use of PCC positively affects patient satisfaction, decreases length of stay, and decreases cost per case [34]. Dementia-based interventions seeking to implement technologies should consider using patient-centered care principles [29,30,35] because PCC includes the caregiver as well as the dementia patient in its target support. If an intervention can help a home-based caregiver prevent a stress-inducing symptom or stress-inducing behavior, both the caregiver and the patient benefit [25,36].

PCC therapies, which are generally non-pharmacological, can improve or maintain quality of life and sometimes improve cognitive function. As shown in Table 4, some non-pharmacologic PCC therapies are easily accessible to home-based caregivers, but some require more specialized efforts. Best practices in dementia care include both education and training for caregivers [30,37].

## 5. Conclusions

As we move into this next generation of the aging population, continued research into the causes, risks, treatments, and cures of dementia is essential. However, research into technological innovation is also critical for the management of the devastating effect of cognitive diseases. Technology offers hope in bridging the gap between needs and available resources. Technology must be human centric; it must address the symptoms and the triggers that cause them. The convergence of technological innovation with a human-centric caregiver can ease anxiety, establish routines, and improve the quality of life for caregivers and dementia patients. Finally, technological innovation is of little use unless we get the technology and resources into the hands of clinical and volunteer caregivers. It is the caregivers and those that they care for who will benefit from easing their burden and improving their quality of life.

## Figures and Tables

**Figure 1 medicina-56-00257-f001:**
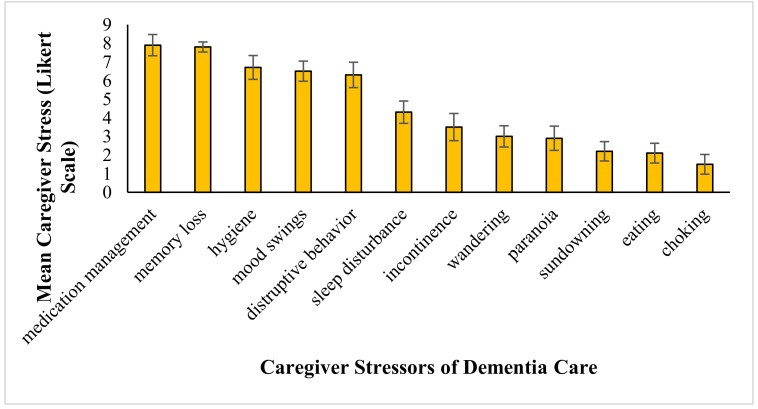
Average caregiver stress (Likert scale) and frequency of behavior or issue. For each common behavior demonstrated by dementia patients, this graph shows how much stress it generates in interviewed caregivers (N = 15) and how common the behavior is. Values for the frequencies of the behaviors come from the literature. Error bars depict standard error of the means.

**Figure 2 medicina-56-00257-f002:**
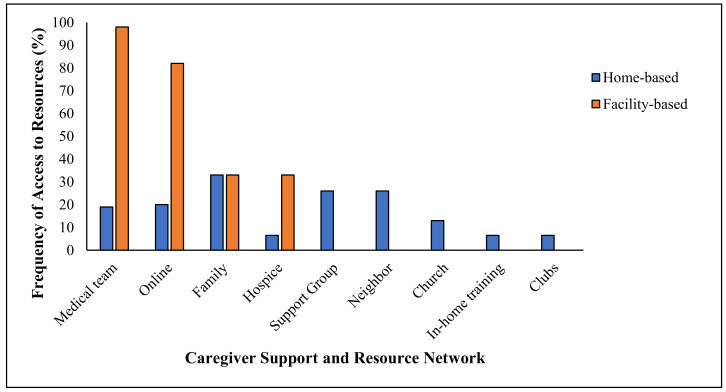
Sources of resources used by home-based caregivers.

**Figure 3 medicina-56-00257-f003:**
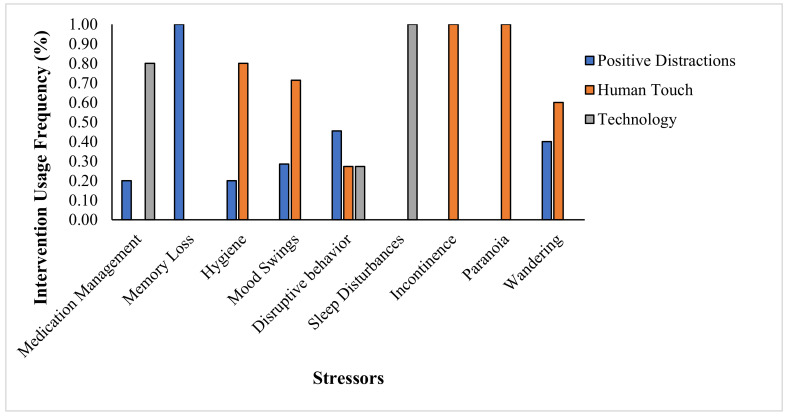
Mitigation methods used by home-based caregivers to manage various dementia-related issues. We asked each home-based caregiver to describe the kind of intervention they used for their three most stress-inducing stressors, as described in Figure 1. Positive distractions include activities such as sharing photo albums or interacting with a pet. Human touch refers to hugs, hand holding, talking, singing and massage. Technology includes television, GPS, or medicine management.

**Figure 4 medicina-56-00257-f004:**
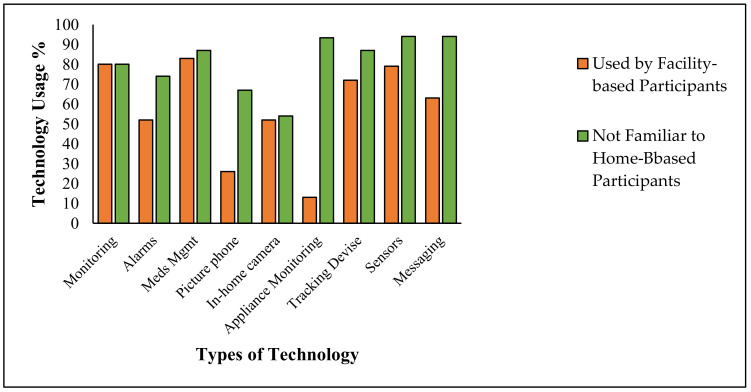
A comparison of the technology used by professional caregivers to those unknown to home-based caregivers. Orange bars represent the technology used by professionals; green bars represent technology with which the home-based volunteers lacked familiarity.

**Table 1 medicina-56-00257-t001:** Description of home-based participants.

Caregiver	Age	Gender	Relationship	Diagnosis	Care Environment
1	82	Male	Wife	Alzheimer’s	In-home with spouse
2	69	Female	Husband	Parkinson’s	In-home with spouse
3	42	Female	Friend	Trauma	Daycare in friend’s home
4	60	Female	Mother	Stroke	In-home with mother
5	70	Male	Wife	Brain tumor	In-home with spouse
6	68	Female	Husband	Alzheimer’s	In-home with spouse
7	60	Female	Mother	Parkinson’s	Daycare in mother’s home
8	55	Female	Mother-in-law	Parkinson’s	Daycare in mother’s home
9	58	Female	Father	Alzheimer’s	In-home care in caregiver’s home
10	64	Female	Husband	Alzheimer’s	In-home care with spouse
11	76	Female	Husband	Alzheimer’s	In-home care with spouse
12	58	Female	Mother	Alzheimer’s	Daycare in mother’s home
13	83	Female	Husband	Vascular dementia	In-home care with spouse
14	74	Female	Husband	PCA dementia	In-home care with spouse
15	80	Female	Husband	Dementia, COPD, diabetes	In-home care with spouse

*Age* and *gender* refer to the participant. *Relationship* describes the person cared for by the participant. *Diagnosis* describes the type of dementia exhibited by the patient. PCA = Posterior cortical atrophy; COPD = Chronic obstructive pulmonary disease. *Care environment* describes where the patient lived and received care.

**Table 2 medicina-56-00257-t002:** Description of the facility-based participants, their patients and environment.

Facility ID	Caregiving Team	Facility Environment	Region	Resident
A	Nurse Mgr, Program director, Env Services, Planetree coordinator, Admin, Physician	2, 16-bedroom memory care households within hospital renovation	East Coast	32 men
B	Nurse Mgr, Program director, Planetree coordinator, Admin, Physician	2, 16-bedroom households, 1 memory care and 1 hospice, new construction, free standing facility	East Coast	32 men
C	Nurse Mgr, Program director, Planetree coordinator, Admin, Physician	2, 16-bedroom households of free-standing new memory care	East Coast =	32 women
D	Nurse Mgr, Program director, Eden Alternative director, Admin, Physician	4, 14-bedroom households of memory care, free standing new construction	Rocky Mt	56 men
E	Nurse Mgr, Program director, Planetree coordinator, Admin, Physician	4, 14-bedroom households of memory care, free standing new construction	East Coast	56 men
F	Nurse Mgr., Admin, Physician	4, 16-bedroom households of memory care, free standing new construction	Midwest	65 residents of mix men and women
G	Nurse Mgr., Admin, Physician	2, 16-bedroom household of memory care within a CCRC new construction	Midwest	34 residents of mix men and women

Seven groups of people (caregiving teams) completed questionnaires. *Caregiving team* describes the composition of the group of people who completed the questionnaire. *Facility environment* refers to the physical structure of the facility. *Region* provides location within the United States, and *resident* refers to the type of patient in each facility.

**Table 3 medicina-56-00257-t003:** Classes of potential stress alleviation and examples as described by the home-based caregivers.

Class of Stress Relief	Example	Speaker
Resources	“Are there any resources that can help with the care, people, products, places? Don’t tell me to call X after the fact… I didn’t know that X even existed.”	Caregiver 8
Information regarding disease progression	“What is it? “How to manage it? What are the symptoms and triggers? What happens next?”	Caregiver 6
Empowerment	“If you are asking me to manage this illness, pay attention to me, I know what I am talking about, I live with it.”	Caregiver 2
Knowledge of costs	“Why is the medical system so complex, I don’t understand the forms and especially what something cost until after they send a bill.”	Caregiver 4
Training	“Help me understand how to use the equipment. Not just the use, but what do I do if something goes wrong?”	Caregiver 3
Knowledge	“I wish I knew about hospice services much earlier. Social workers at the hospital could have discussed this with me when we had the dire diagnosis, right at the beginning. The promotion of hospice should be focused on hospice as a more patient-centered way of managing long-term health care, not just a place where you take your loved one to die. The big point for my wife was to reduce significantly the trips to the hospital.”	Caregiver 5

Open-ended questions allowed caregivers to provide multiple answers.

**Table 4 medicina-56-00257-t004:** Current technology classes and their potential benefit for home-based caregivers and their patients.

Technology	PCC Benefit
Virtual Assistants	Virtual assistants such as smart devices (e.g., Amazon’s Alexa) allow a user to command or ask questions, such that the device then responds to the task. This technology, which can turn on lights and control climates, could address toileting and hygiene issues. Using artificial intelligence, the technology can “learn” to recognize voices and identify better ways to help. New programming could identify innovative purposes and functions. These technologies can lighten the caregiver burden. As these technologies move forward in development, human-centric qualities must be seamlessly integrated [33].
Robots	Robotic technology is growing and promises meaningful solutions for dementia-based technology. Robotic vacuum cleaners use artificial intelligence to assist caregivers and provide entertainment for those who are wheelchair or bed bound [35].
Virtual Reality	Early research has demonstrated a positive experience for those exposed to a virtual reality environment by recalling old memories, reducing aggression and improving their interaction with caregivers. VR has also proved useful in training the caregiver and providing entertainment without leaving the home [33,34].
Music Technologies	The National Institute of Health has awarded $20M to support the first research project of the Sound Health initiative in order to explore the potential of music for treating a wide range of conditions, including dementia. “We hope that these in-depth studies of the science behind music’s influence and impact on the brain will bring real understanding of something we know anecdotally—that music is good for you!” [36].
Internet of Things	Smart technology and the new wave of connected technology can help people with dementia live independently for longer. This new wave of connected technologies, nicknamed “the internet of things,” offers hope for dementia support. Connecting sensors with coordinated care that can collect and process data could help solve the problems of managing the disease and caring for the patient and caregiver [36,37].

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
