# Peer review of "Stress in the Volunteer Caregiver: Human-Centric Technology Can Support Both Caregivers and People with Dementia"

_medicina, 2020, doi:10.3390/medicina56060257_

Round 1
Reviewer 1 Report
In this manuscript, the author explored the use of assistive technologies for dementia care, based on the questionnaire survey and hermeneutical methodology. The author found out the actual status of the use of assistive technologies varied among home-based volunteer caregivers (referred to as A Case) from those in Planetree healthcare personnel (referred to as B Case). The author also investigated the dementia associated behavior (and its triggers) that the caregivers feel stressful to.
This manuscript contains important data about the real condition of dementia care and the possible demand for the new assistive technologies. However, the overall quality of the manuscript does not seem to be high enough to be published in scientific journals. The reviewer thinks this should be because of the rather qualitative method that the author employed in this manuscript. Therefore, the reviewer would like to suggest some revisions to improve the manuscript.
Major points:
- As the author recruited the participants through her personal website, a selection bias can occur which may affect the results and its interpretation. Therefore, it is advisable to make the inclusion criteria (including the period) clearer and discuss the effect of potential selection bias. It is also suggested to show the participants’ characteristics in a table format for A Case (in addition to Table 1), just like in the majority of contemporary clinical studies. The recent research article by Roberts AR et al. (PMID 28977636) could be a good example.
- None of the figures are mentioned in the body of the manuscript. Figures should be referred to in each relevant subsection (https://www.mdpi.com/journal/medicina/instructions)
- As for Figure 2, the author only showed the highest value of the range. Indicating the average value with standard deviation (or standard error) may help the statistical understanding.
- Line 214-296 is too much descriptive. Only introducing the caregivers’ testimony can be more than enough.
- In the Introduction section, the description of line 77-89 (page 2) is significantly overlapping with line 104-110 (page 3). The reviewer thinks the former part can be removed to avoid redundancy. Moreover, the relation between PCC and the need for innovative technology is not clearly discussed, so that the paragraph starting from line 104 gives off an erratic impression. This section needs to be polished.
- Results section should include only the objective outcome obtained from the research. Description in line 190-201 (page 5) is not a result, therefore it should be moved to Introduction or Discussion section. Similarly, line 298-306 (page 8) and 325-336 (page 8) are not the results and should be moved to elsewhere. Line 401-414 (page 11) contains an important discussion, but is not a result.
- Although it is a beginning of the Discussion section, there is no citation of previous research works in line 416-454 (page 11-12). The reviewer recommends the author to read the instruction paper of scientific writing by Hoogenboom BJ et al. 2012 (PMID 23091783).
Minor points:
- The maximum word count for Abstract is 300 words (https://www.mdpi.com/journal/medicina/instructions). So, the document should be slim down and structured with appropriate headings.
- The term “Dementia-based technology” sounds a bit strange. It is suggested to be replaced with more common words.
- As for Figure 1, the maximum value of the y-axis should be 100, not 120.
- If the triggers shown in Table 2 is the data obtained from the interviews, it is better to show somewhat numeric data (e.g. ratio, number of votes, importance).
- In the “Available Technology” column of Table 3, it might be better to cite the information source (publication, textbook, website URL etc.)
There are many other things the reviewer wants to point out, but the reviewer looks forward to see the improved version of the manuscript.
Author Response
Reviewer One Comments:
Major Points
- The Materials and Methods have been revised to address the selection bias. I have included the inclusion criteria for both groups of participants. I have added two new tables that provide background data for home-based caregivers and facility-based caregivers. I have read the suggested clinical study, referenced it and clarified the section as recommended. These changes are addressed in lines 292- 392. The new tables have been added in lines 395-472.
- Figures are now mentioned in the body of the manuscript. They have added in lines:
- 297, 329, 335, 347, 357, 360, 365, 372, 383, 389, 480, 539, 546, 552, 557, 599, 607, 619, 629, 634, 637, 1034, 1036, 1042, 1100, 1109, 1130, 1182, 1209, 1220, 1221, 1245, 1291, 1306, 1319.
- Figure 2 Chart of Dementia Behavior has been changed. The new chart shows Average caregiver stress (Likert Scale) and frequency of behavior. For each common behavior demonstrated by dementia patients, this graph shows how much stress it generates in interviewed caregivers (N = 15) and how common the behavior is. Values for the frequencies of the behaviors come from the literature. Error bars depict standard error of the means.
- The technical description was removed from the section of Dementia Behaviors in lines 536-637. The caregiver comments and minor descriptions remain.
- The Introduction section was rewritten and polished.
- The redundancy was removed. Lines 77-89 were deleted.
- Lines 104-110-Were clarified in the PCC section in regard to DBT
- The revised Introduction 190-287
- As suggested, the Results section was revised as follows:
- Descriptive line items in 190-201 was moved to the Materials and Methods section
- New Material and Methods Section includes the changes on line 292-392
- The description of triggers 298-306 was moved to the Discussion section line 1185-1203
- Lines 325-336 was moved to the end of the Discussion section, lines 1294-1300
- Reference were added on line 1185-[41], line 1203-[27,30], line 1212-[41], line 1222-[41], line 1244-[40].
Minor Points
- Abstract rewritten at 300 words Lines 8-31
- I find Dementia-based technology a suitable description for technology specific to and applicable to dementias. I have defined the term and added a reference in the introduction line 202- 206.
- Revised figure 1 to show the maximum value at 100.
- Numeric data for table 2 now table 4 was not available. The table description is revised to clarify, line 1024-1025.
- In “Available Technology of table 3 now 5 an reference to the Alzheimer’s website was added
Reviewer 2 Report
This manuscripts assess the needs of carers and caregivers of people with dementia, using qualitative approaches. The manuscript is well written and quite well-structured and easy to read.
I have a few comments that may be worth considering:
- The methods section mentioned the number of A Case respondents interviewed but did not mentioned the number of B Case respondents. It becomes apparent later on that there were B Case 6 respondents. This must be mentioned in the methods .
- It's unclear how the percentages assigned to the burden of stress (eg Disruptive behavior 17–83%) obtained. Are these from this particular study or from the literature? Also, the percentage values do not match those shown in Figure 2
- There were a few formatting issues that require fixing, for example table 2 and table 3.
Overall, a very important work which is providing useful knowledge about how technology could be used at various stages of dementia as the disease progresses.
Author Response
Review Two:
- New charts describing A Case respondents and B Case Respondents were added. have included the inclusion criteria for both groups of participants. I have added two new tables that provide background data for home-based caregivers and facility-based caregivers. These changes are addressed in lines 292- 392. The new tables have been added in lines 395-472.
- Figure 2 Chart of Dementia Behavior has been changed. The new chart shows Average caregiver stress (Likert Scale) and frequency of behavior. For each common behavior demonstrated by dementia patients, this graph shows how much stress it generates in interviewed caregivers (N = 15) and how common the behavior is. Values for the frequencies of the behaviors come from the literature. Error bars depict standard error of the means.
- Formatting and description in Table 2 now table 4 has been revised. Likewise with Table 5.
Other revisions:
- Minor editing lines 281, 1122, 1143-1174, 1307-1310
- Add acknowledgements
- Revised and reformatted references
- Added new references, 41,4243,44 & 45
Round 2
Reviewer 1 Report
In this revised manuscript, the author addressed some of the concerns that the reviewer raised for the initial manuscript. It seems the manuscript was improved to some extent. However, the way of the author’s writing is so unique that the reviewer finds it difficult to imagine the best form of this article.
The reviewer truly wants another peer reviewer’s opinion. In addition, the reviewer is wondering if Dr. Sharon Pochron or Dr. Thomas H. Wan, who is credited in the Acknowledge section, can be a co-author of this manuscript to support the author by exchanging views.
What the reviewer can point out at this point is as follows:
- Abstract should be a summary of the entire manuscript. However, it looks like repeating the same thing for the Introduction section. Since this is not a review article but an original investigation, more structured abstract would be appreciated. For example, one can start the first 50 words for describing the background, followed by the next 200 words for showing methods and results, and then use the last 50 words for the conclusive remarks.
- As for Figure 1, the reviewer does not fully understand what “A Case Unknown” (orange color bar) really stands for. If it means the ratio of people who is unfamiliar to each technology, it should not be included in the same graph. It sounds better to just include the user ratio of “A Case” and “B Case” and assess the difference by using Chi-square analysis. The data of “A Case Unknown” can be just described in the manuscript (i.e. “The most unfamiliar technology for participants of A Case was in-home camera (8/15, 53%), followed by picture phone (5/15, 33%) and so on) or a new table separated from Figure 1.
- Figure 2 has been improved by indicating values with mean ± However, it is still misleading because it looks as if it is comparing between blue bars and yellow bars. It is advisable to separate Figure 2 into two panels (i.e. Panel A for the Likert Scale and panel B for the frequency).
- Considering the aim of Figure 3, pie charts can be better to highlight the characteristics of Case A and Case B, rather than using bar graphs.
The reviewer really hope this peer-review can contribute to improve the quality of manuscript.
Author Response
May 8, 2020
MDPI, Medicina Journal
RE: Manuscript Article Revision
Dear Editor:
I am submitting this revision to my research article 757240-8 “Stress in the Volunteer Caregiver: Human-Centric Technology can Support Caregivers and People with Dementia” to your journal Medicina for the special issue, “Artificial Intelligence Research in Healthcare” under the invitation from Dr. Thomas TH. Wan, special issue editor.
As requested by reviewer 1, Sharon Pochron, PhD, has been added as a co-author to this manuscript to support the author. Dr Pochron is well published in peer review journal within multiple scientific fields. The co-author has revised the entire manuscript including all issues brought up ty the reviewers.
Reviewer One Comments:
- Abstract has been rewritten.
- Figure 1 has been revised. A Case has been changes to “home-based caregiver” and B Case has been changed to “facility-based caregiver” for clarification.
- Figure 2 has been revised to improve meaning.
- Figure 3 is a new chart.
Other revisions:
- Title has been revised to remove jargon as recommended by the co-author
- Table 3 has been added to clarify the Results.
- Figure 4 previously Figure 3 has been revised.
- Table 5 is a new table to support results
- Materials and Methods have been edited extensively
- The Discussion section has been rewritten.
- Revised and reformatted references
I would like to thank the reviewers for their comments and feel their recommendations have greatly improved the manuscript.
Thank you
Barbara Huelat, FASID, EDAC, AAHID